# Genetic Parameters and Inbreeding Effect of Morphological Traits in Sardinian Anglo Arab Horse

**DOI:** 10.3390/ani10050791

**Published:** 2020-05-02

**Authors:** Andrea Giontella, Francesca Maria Sarti, Giovanni Paolo Biggio, Samira Giovannini, Raffaele Cherchi, Camillo Pieramati, Maurizio Silvestrelli

**Affiliations:** 1Department of Veterinary Medicine—Sportive Horse Research Center, University of Perugia, via S.Costanzo 4, 06123 Perugia, Italy; camillo.pieramati@unipg.it (C.P.); maurizio.silvestrelli@unipg.it (M.S.); 2Department of Agricultural, Food and Environmental Sciences, University of Perugia, Borgo XX Giugno, 74, 06121 Perugia, Italy; francesca.sarti@unipg.it (F.M.S.); samira.giovannini@gmail.com (S.G.); 3AGRIS, Servizio Ricerca Qualità e Valorizzazione delle Produzioni Equine, piazza D. Borgia, 4, Ozieri, 07014 Sassari, Italy; giampaolo.biggio@gmail.com (G.P.B.); ilex283@gmail.com (R.C.)

**Keywords:** Sardinian Anglo-Arab horse, morphology, inbreeding, genetic parameters

## Abstract

**Simple Summary:**

Cannon bone circumference, chest girth, shoulder length, and withers height are important biometric measures that are useful to describe the horse’s body conformation correctly. It is interesting to observe how these traits are connected to each other and how they are used to improve the genetic progress of the breed. The percentages of Thoroughbred and Arabian blood were also estimated for each horse in the pedigree population, to evaluate how the conformation might be modified thanks to different breeding purposes such as: racing, jumping, and other equestrian disciplines. The results show that the improvement of the Arabian blood determined a depressive effect on the studied traits; on the contrary, when the Thoroughbred blood percentage increases, the measures of the studied traits increase too. The trait most affected by the variation of the two blood percentages was withers height. It has been demonstrated that the conformation is related to sporting results, soundness, and durability and that body and legs shape are highly heritable. At this purpose, it might be useful, to restart the body measurements collection and to match them with the horse’s performance at races.

**Abstract:**

The purpose of this study was to estimate the heritability and genetic correlations of four biometric measurements and an overall score (OS) in the Sardinian Anglo-Arab horse (SAA); moreover, the effect of inbreeding on these traits was investigated. A dataset with 43,624 horses (27,052 females and 16,572 males) was provided by the Agricultural Research Agency of Sardinia (AGRIS). Cannon bone circumference (BC), chest girth (CG), shoulder length (SL), and withers height (WH) were measured on 6033 SAA horses born in Sardinia between 1967 and 2005; beside the measurements, an overall score (OS) was taken comparing the morphology of each horse to an “ideal type” that is scored out of 100. The mean value is 20.5 cm for BC, 185.9 cm for CG, 67.6 cm for SL, 160.8 cm for WH, and 73.2 for the OS. The heritability estimates ranged from 0.78 to 0.23. The results allow to foresee high genetic progress through the breeding programs. The most affected trait by the inbreeding rate seems to only be the withers height.

## 1. Introduction

The Sardinian Anglo-Arab (SAA) is a very popular horse for equestrian sports; it originated from crosses between indigenous mares (old Arabian lines and Part-Arab) with Arab and Thoroughbred stallions; its breeding and selection started in the ancient ages [1,2].

The first relevant breeding farm in Sardinia was Regia Tanca of Paulilatino; it originated between VIII-XI century A.C.; in the Aragonese-Spanish period (1412–1720), Regia Tanca started a breeding program which included morphological and functional traits. In the 1800s, under the Carlo Felice’s reign, Regia Tanca became the public Remount Station in Ozieri (50 km S-E from Sassari) which, for the first time, made it possible to use stallions born in the royal stables of Turin and Tripoli.

The selection aimed to achieve different goals such as: racing, jumping, and other equestrian disciplines [3,4].

In 1967, the administrative board of the “Istituto Incremento Ippico della Sardegna” defined the Sardinian Anglo-Arab as: “the product of the selection and crossbreeding between Arabian lines, Thoroughbred, Anglo-Arab Thoroughbred, Sardinian Anglo-Arab stallions and indigenous mares with a percentage of Arabian blood not less than 25% and not more than 75%”.

Moreover, it has to be pointed out that the use of the Thoroughbred stallions was not encouraged until the sixties of the last century, yet the good results obtained through this cross led many breeders to utilize this breed that therefore spread all over the population.

The Italian Studbook of SAA was established in 1967, and in 2004, the National Association of Anglo-Arab horse breeders (A.N.A.C.A.A.D.) was born.

In this study, four traits (cannon bone circumference—BC, chest girth—CG, shoulder length—SL, withers height—WH) were measured in SAA horses born in Sardinia between 1967 and 2005. Withers height is used to classify the SAA in small, medium and large types [3].

The four measurements were collected when the horses were at least three years old and participated in specific shows organized by A.N.A.C.A.A.D. in Sardinia. During the show, a scientific committee was used to judge male and female horses through a morphological evaluation by an overall score (OS) that compared their morphology and gaits to an “ideal type” scored 100. It must be specified that the SAA Studbook is open; a horse is suitable to be enrolled in the Studbook if at least two out of three mandatory conditions are respected: an OS ≥ 50, good race results, parents enrolled in the Studbook. For the traits’ measurement, the minimum values allowed are: 17 cm for BC, 157 cm for CG, 58 cm for SL and 140 cm WH.

Compared to other horse breeds, the SAA selection plan seems very old-fashioned, although it has to be pointed out, that judging the conformation by comparison to an “ideal type” has a long-standing tradition in horse breeding; in fact, the conformation largely determines the general appearance of the horse. Moreover, it has been demonstrated that the conformation is related to sporting results, soundness and durability [5] and that body and legs shape are highly heritable [6,7,8].

To avoid the subjectivity in the comparison to an ideal type in many species, the overall score has been supplemented or replaced by several simple traits, which are linearly evaluated on a scale from one biological extreme to the other [9,10]; unlike the previous approach, the linear evaluation does not grade an animal, but rather describes it. Although the linear evaluation is now adopted for some horse breeds [11], few traits of the horse’s body are considered so important that they are actually measured and not evaluated on a linear scale [8,12]. At the present time, in the SAA, the linear evaluation has not yet been applied, but it is tested in a random sample of the population in order to evaluate its practical application.

Taking into account that the SAA breed originated from crossbreeding and that the selection was planned to achieve different goals during its history, this study defines the genetic determination of the traits involved in its breeding program.

## 2. Material and Methods

A database with BC, CG, SL, WH and OS measures of 6033 SAA born from 1967 to 2005 was provided by Agricultural Research Agency of Sardinia - Istituto Incremento Ippico della Sardegna (AGRIS); after this year, the Agency stopped collecting these data because it was too expensive. The four body measurements were directly obtained from the left side of the horse using Lydthin stick and tape measure. BC, the cannon bone circumference, was measured with tape, CG the heart girth circumference, was detected with tape behind the shoulders, SL from the point of shoulder (under the supraglenoid tubercle) to scapula dorsal border, detected with tape, WH the withers height, were measured with a Lydthin stick from the withers to the floor. (Figure 1)

Moreover, a pedigree file including 43,624 horses (27,052 females and 16,572 males) registered in the Studbook kept by the Ministry of Agricultural Policies was provided by AGRIS. This pedigree file was used to set up the BLUP animal models. The statistical significance of measures and OS between males and females were estimated with a general linear model, which takes into consideration the fixed effect of the sex by the R Core Team (2020) [13].

A single trait animal model, including the fixed effects of sex and of year of birth, was run to estimate variance components for each morphological trait (data not shown). The single trait model estimates were used as priors for a multiple trait (MT) animal model with five traits (BC, CG, SL, WH, OS); the MT model included the same fixed effects as the single trait models.

Variance components estimation and their standard errors were performed by VCE package [14]; the estimates of the EBVs and their accuracies (r_TI_) were calculated by MTDFREML [15].

The percentages of Thoroughbred and Arabian blood were also estimated for each horse in the pedigree population (43,624 horses). The first one was calculated using in-house Fortran 95 software [16,17]; the percentage of Arabian blood was available in the AGRIS dataset and officially certified by the breeder association; the Arabian blood percentages were, however, recalculated by the same in-house software to check the reliability of the official data. Through the in-house software, the inbreeding coefficients (F), the correlation between Thoroughbred and Arabian blood percentages and morphological traits, were also calculated.

## 3. Results

The overall mean was 20.5 cm for BC, 185.9 cm for CG, 67.6 cm for SL, 160.8 cm for WH, and 73.2 for the OS (Table 1).

As expected, all of the measurements and the overall score were significantly higher in males. In general, coefficients of variation (not tabulated data) were from moderate to low (3.8 to 7.0 in males and 2.6 to 6.4 in females), showing a highly homogeneous population.

The heritability estimation of the morphological traits in the MT model (Table 2) were the following: 0.57 (CB), 0.47 (CG), 0.34 (SL) and 0.78 (WH).

The genetic correlations were all positive and high, ranging from 0.71 (WH and CG) to 0.86 (WH and SL); on the contrary, the environmental correlations showed lower values, with a minimum of 0.21 (BC and SL) and a maximum of 0.40 (CG and SL). The heritability of OS was low (0.23) and the correlations of OS with the morphological traits were also low: the genetic correlations ranged from 0.23 (WH) to 0.42 (SL), the lowest environmental correlation was 0.17 (BC) and the highest 0.27 (WH).

The trend of the genetic indexes for the studied traits by year of birth is shown in Figure 2.

The similarity of the measurement patterns of the morphological traits is due to their high genetic correlations; the differences observed with OS before 1995 are probably caused by the very low number of horses (only stallions) evaluated for this character in that period. The accuracies of the indexes ranged from 0.66 (OS) to 0.90 (WH).

The percentages of Arabian and Thoroughbred blood by year of birth are shown in Figure 3: the percentage of the Arabian blood was higher, until 1974, when the percentage of the Thoroughbred blood became prevalent.

Between 1974 and 1989, both percentages increased; however, the Thoroughbred was only slightly higher. After 1989, the Arabian blood became constant, while the Thoroughbred increased, even if only slightly. Only few differences were found between Arabian blood percentages by AGRIS dataset and the recalculated percentages: the correlation was 0.96; most differences were due to rounding, although mistyping was also found.

In Table 3, the correlations between the morphology and the inbreeding coefficient, the Arabian blood percentage and Thoroughbred blood percentage, are reported.

Regarding the inbreeding, it must be outlined that apart from SL, all traits have a negative correlation with this parameter, although the significance level is reached only by WH (−0.042).

A different situation is observed for the Arabian and Thoroughbred blood percentages; only in one case (OS-Arabian blood) is the correlation not significant (−0.017).

A general consideration demonstrates that the improvement of the Arabian blood determined a depressive effect on the studied traits; on the contrary, when the Thoroughbred blood percentage increases, the measures of the studied characters also improve. The trait most affected by the variation of the two blood percentages was WH.

## 4. Discussion

The SAA is characterized by solid feet and legs, a refined head with a straight profile, and a fiery temperament with speed and stamina. According to the wither heights, there are 3 types in the breed: small type (156–159 cm), medium type (160–165 cm) and large type (taller than 165 cm); it has to be noted that the threshold between small and medium type was 158 cm until 1984 [3]. The percentage of small, medium and large types in the sample were 32.7%, 49.2% and 18.1%, respectively.

In the last century, during the 1930s, the WH of SAA was 150 cm; in 1971–1972, it increased to 161 cm and it subsequently increased to 163 cm in 1983–1984 [3]. Surprisingly, in 1985–2005, the WH mean decreased to 161 cm; during the same period, the inbreeding coefficient increased and a significant negative correlation between WH and F was observed (Table 3). SL, CG, BC and OS also showed this trend, although no significant correlations with F were found for these traits.

During 1971–1972, the SL mean was 65.5 cm; in 1983–1984, it increased to 67.0 cm [3] to reach 67.9 cm in 1985–2005. It is known that a long shoulder provides a greater base for the long muscles useful for upward movement, flat racing and other movement traits [18]. In the beginning, this change was probably not caused by a selection for sports, but by breeders choosing long shouldered animals to obtain sport type horses. The measures of CG and BC showed the same growing trend until 1984: it was reported that CG improved from 178.5 to 188 cm and BC from 19.7 to 20 cm [3]; after that period, these measurements decreased: CG dropped to 186.2 cm and BC to 19.8 cm.

It was observed that as the Arabian blood percentage increased, the horse size got smaller, as shown by the negative significant (*p* < 0.01) correlations with the body traits; on the contrary, the same traits are positively and significantly (*p* < 0.01) correlated with Thoroughbred blood. This situation can be the result of the breeding program implemented for this breed. Moreover, it has to be taken into consideration that Thoroughbred was widely used in many local breeds to obtain a less bulky and lighter horse, more suitable for several sports [18,19]: the percentage of Thoroughbred blood in SAA greatly increased in 1985–1990 and slightly increased thereafter.

A different situation is observed in OS; this trait seems to be rather independent from the Arabian and Thoroughbred blood percentages, as highlighted by the very small correlations in Table 3 (0.017 ± 0.014 ns and 0.060 ± 0.013, *p* ≤ 0.01).

Inbreeding had a significant effect on reducing WH (−0.042), but it had no effect on other morphological traits. A higher, but not significant, effect of inbreeding was estimated by Gomez et al. [20] on Spanish horse, and moreover, a significant inbreeding effect is reported by Curik et al. [21] on Lipizzan horse, Gandini et al. [22] on Italian Haflinger, Oom [23] on Lusitano horses and Sierszchulski et al. [24] on Arab males.

The heritability estimates of these biometric measures and overall score in other breeds laid in a medium to high range. The differences between h^2^ reflected the practical difficulty in taking the measure of the traits; the heritability estimation of the OS was rather low (0.22 ± 0.02). This fact could be due not only to the composite nature of the trait, but also to its heavy dependence on the subjectivity of the judge whose name, unfortunately, was not available, so that this effect could not be included in the statistical models.

The h^2^ of WH (0.78 ± 0.02) was similar to the h^2^ in Pura Raza Española (0.80 ± 0.03) [8], although it was higher than the h^2^ reported by other authors in different breeds. In the Arab breed, it was equal to 0.48 [25]; in the Wielkopolsky, it was 0.57 ± 0.01 [19] and, finally, in Pantaneiro breed, it increased to 0.61 [6].

The estimate of h^2^ for CG (0.47 ± 0.02) put this trait among those with medium-high heritability as well as reported in other studies. In the Arab horse, this value was 0.31 ± 0.01 [25] and in the Wielkopolsky, it was equal to 0.35 ± 0.01 [19]. Only in the Pura Raza Española was the CG heritability very high, reaching 0.67 ± 0.03 [8]. For the SL trait, the h^2^ value found in this study was comparable with that of the Belgian warmblood, which was equal to 0.31 [7]; in the Pura Raza Española, it was 0.55 ± 0.03 [8] and 0.69 in the Pantaneiro horse [6]. The heritability estimation of the BC trait was equal to 0.57 ± 0.02 and it was higher than the value 0.44 ± 0.03 computed on the Pura Raza Española [8] and on Wielkopolsky (0.42 ± 0.01) [19]. The value of h^2^ in SAA was closer to the 0.51 obtained for the Arabian horse [24] and equal to the value of h^2^ in the Friesian Horse population of South Africa and Namibia [11].

Comparing the h^2^ of SAA measurements with the values of Murgese, another Italian horse breed, it is possible to observe marked differences: the heritability estimations for the three traits WH, CG and BC were lower and respectively 0.27 (vs. 0.78 SAA), 0.31 (vs. 0.47 SAA) and 0.24 (vs. 0.57 SAA). It must be highlighted that the Murgese population is much smaller than SAA [26].

The genetic correlations (Table 3) between the morphological measurements in SAA were very similar to those estimated in Murgese (WH-CG 0.71 vs. 0.707, WH-BC 0.73 vs. 0.64 and CG-BC 0.74 vs. 0.709) [26]. The same genetic correlations in SAA were slightly higher than those reported in other breeds such as: 0.55 in Wielcopolsky horse [19], 0.40 in Pura Raza Español [8] and close to the values (0.76–0.84) estimated on Finnhorse trotter by Saastamoinen et al. [27]

The h^2^ estimate for OS (0.23 ± 0.02) was very difficult to compare with other studies: in fact, it was an overall subjectively scored “ideal type” and there were many differences in the way of scoring, in the judged components and in the breeding goal [28].

## 5. Conclusions

The obtained results highlight that the morphological feature of SAA is influenced in a rather balanced way by the two founder breeds; in fact, significant negative correlations (*p* < 0.01) between all morphological measures with Arabian blood percentage, as well as positive correlation (*p* < 0.01) with Thoroughbred blood, were found. The breeding program implemented during the years for the body measurements could explain this situation. Because the correlations between the OS and Arabian and Thoroughbred blood percentages are very small, we can hypothesize that trait is independent from the breed composition.

In SAA, body traits were highly heritable as in other breeds; this permits one to foresee a high genetic progress through the breeding programs. For this reason, it needs to be highlighted that it is necessary to restart the collection of body measurements. A good correlation between the studied traits was also estimated. These results demonstrate that it would be interesting to match both the morphological and performance data in the future.

## Figures and Tables

**Figure 1 animals-10-00791-f001:**
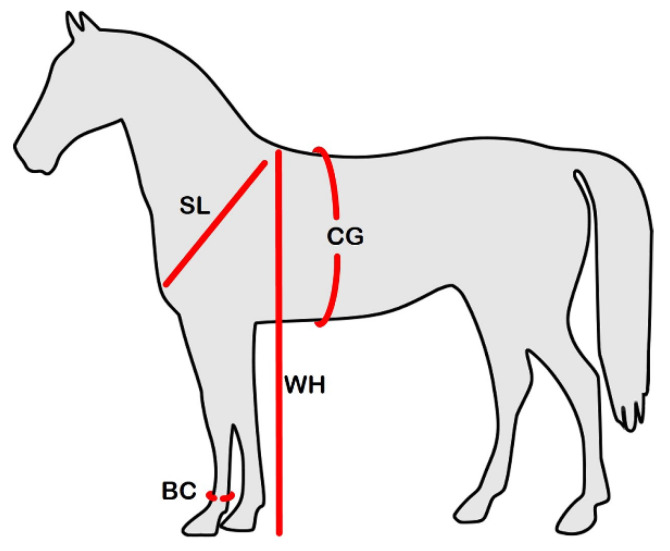
The four body measurements detected in Sardinian Anglo-Arab horse (SAA). BC = Cannon Bone Circumference; CG = Chest Girth; SL = Shoulder Length; WH = Withers Height.

**Figure 2 animals-10-00791-f002:**
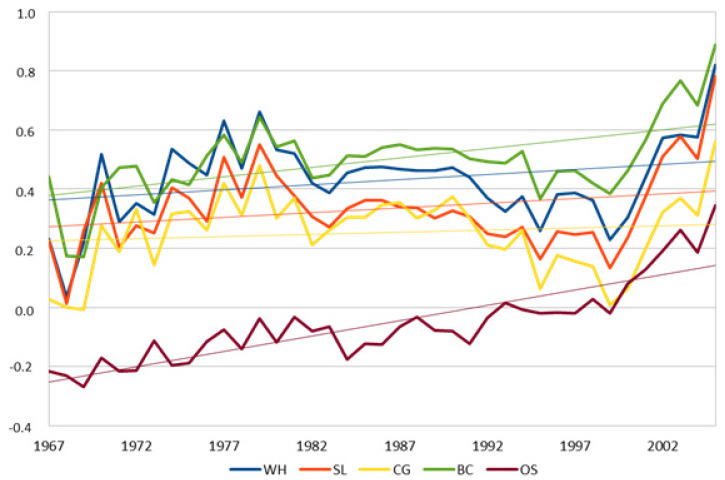
The trend of genetic indexes in the studied traits by year of birth.

**Figure 3 animals-10-00791-f003:**
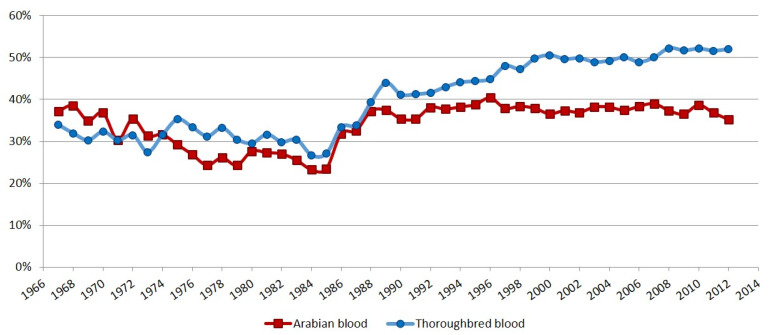
The percentages of Arabian and Thoroughbred blood by year of birth.

**Table 1 animals-10-00791-t001:** Mean ± s.d. of body traits (cm) and overall score (out of 100).

	BC	CG	SL	WH	OS	n
**males**	20.9 ± 1.1 *	188.5 ± 13.2 *	68.8 ± 2.8 *	162.9 ± 6.2 *	74.2 ± 2.4 *	1537
**females**	20.1 ± 1.2 *	185.6 ± 8.3 *	67.6 ± 4.3 *	160.7 ± 4.2 *	73.2 ± 3.6 *	4496
**overall**	20.5 ± 1.0 *	185.9 ± 7.0 *	67.6 ± 2.7 *	160.8 ± 4.3 *	73.2 ± 3.5 *	

BC = Cannon Bone Circumference; CG = Chest Girth; SL = Shoulder Length; WH = Withers Height; OS = Overall Score; *: *p* < 0.05.

**Table 2 animals-10-00791-t002:** Estimates of heritability ± s.e. (diagonal), genetic (above diagonal) and environmental (below diagonal) correlation ± s.e. of the final five traits model.

Trait	BC	CG	SL	WH	OS
BC	**0.57 ± 0.02**	0.74 ± 0.03	0.73 ± 0.03	0.73 ± 0.02	0.29 ± 0.03
CG	0.33 ± 0.02	**0.47 ± 0.02**	0.77 ± 0.02	0.71 ± 0.02	0.37 ± 0.03
SL	0.21 ± 0.02	0.40 ± 0.02	**0.35 ± 0.02**	0.86 ± 0.02	0.42 ± 0.05
WH	0.25 ± 0.03	0.24 ± 0.03	0.33 ± 0.02	**0.78 ± 0.02**	0.23 ± 0.04
OS	0.17 ± 0.02	0.21 ± 0.01	0.22 ± 0.02	0.27 ± 0.03	**0.23 ± 0.02**

BC = Cannon Bone Circumference; CG = Chest Girth; SL = Shoulder Length; WH = Withers Height; OS = Overall Score.

**Table 3 animals-10-00791-t003:** Linear correlation coefficient ± s.d. of body traits with inbreeding coefficient, Arabian blood percentage and Thoroughbred blood percentage.

	Inbreeding	Arabian Blood	Thoroughbred Blood
BC	−0.005 ± 0.013	−0.304 ± 0.012 **	0.129 ± 0.013 **
CG	−0.010 ± 0.013	−0.307 ± 0.012 **	0.136 ± 0.013 **
SL	0.005 ± 0.013	−0.257 ± 0.012 **	0.168 ± 0.013 **
WH	−0.042 ± 0.013 *	−0.442 ± 0.012 **	0.301 ± 0.012 **
OS	−0.024 ± 0.014	−0.017 ± 0.014	0.060 ± 0.013 **

BC = Cannon Bone Circumference; CG = Chest Girth; SL = Shoulder Length; WH = Withers Height; OS = Overall Score; *: *p* < 0.05; **: *p* < 0.01.

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
