# Peer review of "Genetic Parameters and Inbreeding Effect of Morphological Traits in Sardinian Anglo Arab Horse"

_animals, 2020, doi:10.3390/ani10050791_

Round 1

Reviewer 1 Report

The submitted paper is a notable documentation of the genetic component for several straightforward morphological measures in an out-crossed breed of horse.  The scientific design is straightforward and appropriate.  I am particularly excited to see a well documented case of inbreeding depression on withers height.  However, the manuscript must be extensively edited for grammar and clarity.  I strongly suggest the authors employ a professional English language editing service.I've made some suggestions, but the changes needed were far too extensive to document completely in this review.

Line 21, perhaps exchange the word “improve” for increase. There are those who don’t think bigger is better….

Line 22, please define the abbreviation WH the first time it is used.  The last sentence is also a bit long and awkward and could be revised.

Line 35, the last sentence of the abstract is very rough.  Please revise…

Please strive to reduce your use of the semi-colon (;).  In most cases these are two ideas better expressed in two separate sentences.  This occurs in so many places in the text that I cannot note them all.

Line 43, delete “the one of” and, it’s originating, not “originated”

There are many paragraphs with just one sentence. This is not considered good grammar, and leaves the reader feeling a bit jostled.  Strive to improve the flow and organization throughout, especially in the background.

Line 65, perhaps “compared to other [agricultural] species”, this was quite confusing as it is currently written. 

Line 73, I disagree that linear evaluation is now “usual in the horse”. Perhaps in Europe, but not globally.  It should be, but we are a long way off still.

Methods:

The software packages used are a bit dated, but as these methods have changed little, this is tolerable.  The “in-house” code should be documented or referenced in some manner.  Today’s standard would probably be deposition in GitHub.  

Table 1: the column headings have gotten out of alignment in each of the tables.

Please run all statistics separating our intact males from geldings.  Many studies have shown changes in growth as a result of castration and this variable should be a key part of your model.

Line 111: you say “as expected, all measures were significantly higher in males” and yet… you did not give us a statistic to describe this difference.  Please provide a statistical test documenting differences between sexes across all years (and, as above, please split out stallions vs. geldings.)

Figure 2: Is this the mean % pedigree ancestry by year?  Could we see the % of unknown or local horse ancestry as well?  Please present each data point as a mean +/- s.d.

Line 161: delete “the”, and change “for” to from.

Line 184: change “suited” to type

The abbreviation for heritability should be written with the 2 in superscript throughout (rather than “h2”).

Line 225: change “really” to much

Reviewer 2 Report

The article presents the analysis of genetic parameters and inbreeding effect of morphological features in a Sardinian Anglo-Arab horse (SAA).

Four parameters were used for the analysis: cannon bone circumference (BC), chest girth (CG), sholder lenght (SL) and withers height (WH). No exact measurement methodology has been given (anatomical points, especially for SL) The heritability and genetic correlations of four biometric measurements and the overall score (OS) in a Sardinian Anglo-Arab horse were estimated based on measurements of 6033 SAA horses born in Sardinia in 1967-2005.Beside the measurements, an overall score (OS) was taken comparing the morphology of each horse to a "ideal type" that is scored 100. Pedigree analysis to determine the effect of inbreeding on these traits was performed on the data of 43 624 horses (27 052 females and 16 572 males). Taking into account the number of horses measured - 6033 (no data on the number of stallions and mares), the obtained test results and formulated conclusions were developed on approximately 14% of the population, in the case of genetic parameters, it is noted whether such a small representative group can draw conclusions for the entire population. There were no minimum values of the tested parameters to be entered in the studbook. The calculated average value of OS 73,2 seems to be low in relation to 100 scale. There is no reference to the minimum required. Generally, three basic measurements of cannon bone circumference (BC), chest girth (CG) and withers height (WH) are given in the studbook entery for horses of different breeds. The work uses data from 15 years ago, which dose not given a picture of the modern Sardinian Anglo-Arab horse population. The work is of historical nature for possible comparision when assessing the modern population of the SAA horse. The tables have bad descriptions and no data, e.g. "n" in Table 1.

Round 2

Reviewer 1 Report

The responses to the first review have addressed most concerns adequately.  Thanks for seeking out English language editing.

The only remaining comment is that the reference given for the Fortran software in the response to review should be added to the manuscript itself.

Author Response

The only remaining comment is that the reference given for the Fortran software in the response to review should be added to the manuscript itself.

L18-19: we thank the Referee for the remark. As suggested we added the two references in the Material and Method section.

Reviewer 2 Report

I believe the manuscript has been significantly improved and now warrants publication in Animals.

Author Response

We thank the Referee for the positive comment.